# Method for identification of condition-associated public antigen receptor sequences

**Mikhail V Pogorelyy[1], Anastasia A Minervina[1], Dmitriy M Chudakov[1,2,3], Ilgar Z Mamedov[1], Yuri B Lebedev[1,4†*], Thierry Mora[5†*], Aleksandra M Walczak[6†*]**

[1]Department of Genomics of Adaptive Immunity, Shemyakin-Ovchinnikov Institute of Bioorganic Chemistry of the Russian Academy of Sciences, Moscow, Russia; [2]Center for Data-Intensive Biomedicine and Biotechnology, Skolkovo Institute of Science and Technology, Moscow, Russia; [3]Central European Institute of Technology, Brno, Czech republic; [4]Biological Faculty, Moscow State University, Moscow, Russia; [5]Laboratoire de Physique Statistique, CNRS, Sorbonne University, Paris-Diderot University, École Normale Supérieure, Paris, France; [6]Laboratoire de Physique Theorique, CNRS, Sorbonne University, École Normale Supérieure, Paris, France

**\*For correspondence:**
lebedev_yb@mx.ibch.ru (YBL);
tmora@lps.ens.fr (TM);
awalczak@lpt.ens.fr (AMW)

†These authors contributed equally to this work

**Abstract** Diverse repertoires of hypervariable immunoglobulin receptors (TCR and BCR) recognize antigens in the adaptive immune system. The development of immunoglobulin receptor repertoire sequencing methods makes it possible to perform repertoire-wide disease association studies of antigen receptor sequences. We developed a statistical framework for associating receptors to disease from only a small cohort of patients, with no need for a control cohort. Our method successfully identifies previously validated Cytomegalovirus and type one diabetes responsive TCR$\beta$ sequences .
DOI: https://doi.org/10.7554/eLife.33050.001

## Introduction

T-cell receptors (TCR) and B-cell receptors (BCR) are hypervariable immunoglobulins that play a key role in recognizing antigens in the vertebrate immune system. TCR and BCR are formed in the stochastic process of V(D)J recombination, creating a diverse sequence repertoire. These receptors consist of two hypervariable chains, the $\alpha$ and $\beta$ chains in the case of TCR. Progress in high throughput sequencing now allows for deep profiling of TCR$\alpha$ and TCR$\beta$ chain repertoires, by establishing a near-complete list of unique receptor chain sequences, or 'clonotypes', present in a sample. Most sequencing data available correspond to TCR$\beta$ only, but the same principles discussed below apply to TCR$\alpha$ repertoires, or to paired $\alpha\beta$ repertoires.

Comparison of sequenced repertoires has revealed that in any pair of individuals, large numbers of TCR$\beta$ sequences have the same amino acid sequence (*Venturi et al., 2011*). Several mechanisms leading to the repertoire overlap have been identified so far. The first mechanism is *convergent recombination*. Due to biases in V(D)J recombination process, the probability of generation of some TCR$\beta$ sequences is very high, making them appear in almost every individual multiple times and repeatedly sampled in repertoire profiling experiments (*Britanova et al., 2014*). This sharing does not result from a common specificity or function of T-cells corresponding to the shared TCR$\beta$ clonotypes, and may in fact correspond to cells from the naive compartment in both donors

(*Quigley et al., 2010*), or from functionally distinct subsets such as CD4 and CD8 T-cells. The second possible reason for TCR sequence sharing is specific to identical twins, who may share T cell clones as a consequence of cord blood exchange *in utero* via a shared placenta (*Pogorelyy et al., 2017*). Note that in that scenario both the $\beta$ and $\alpha$ chains are shared together. The third and most interesting mechanism for sharing the sequence of either the $\beta$ or $\alpha$ or both chains is *convergent selection* in response to a common antigen. From functional studies, such as sequencing of MHC-multimer specific T-cells, it is known that the antigen-specific repertoire is often biased, and the same antigen-specific TCR $\beta$ or $\alpha$ chain sequences can be found in different individuals (*Miles et al., 2011*; *Dash et al., 2017*; *Glanville et al., 2017*).

Reproducibility of a portion of the antigen-specific T-cell repertoire in different patients creates an opportunity for disease association studies using TCR$\beta$ repertoire datasets (*Faham et al., 2017*; *Emerson et al., 2017*). These studies analyse the TCR$\beta$ sequence overlap in large cohorts of healthy controls and patients to identify shared sequences overrepresented in the patient cohort. Here we propose a novel computational method to identify clonotypes which are likely to be shared because of selection for their response to a common antigen, instead of convergent recombination. Our approach is based on a mechanistic model of TCR recombination and is applicable to small cohorts of patients, without the need for a healthy control cohort.

## Results

As a proof of concept, we applied our method to two large publicly available TCR$\beta$ datasets from Cytomegalovirus (CMV)-positive (*Emerson et al., 2017*) and type one diabetes (T1D) (*Seay et al., 2016*) patients. In both studies the authors found shared public TCR$\beta$ clonotypes that are specific to CMV-peptides or self-peptides, respectively. Specificity of these clonotypes was defined using MHC-multimers. We show that TCR$\beta$ chain sequences functionally associated with CMV and T1D in these studies are identified as outliers by our method.

The main ingredient of our approach is to estimate the probability of generation of shared clonotypes, and to use this probability to determine the source of sharing (see *Figure 1*). Due to the limited sampling depth of any TCR sequencing experiment, chances to sample the same TCR$\beta$ clonotype twice are low, unless this clonotype is easy to generate convergently, with many independent generation events with the same TCR$\beta$ amino acid sequence in each individual (convergent recombination), or if corresponding T-cell clone underwent clonal expansion, making its concentration in blood high (convergent selection). Thus, we reasoned that convergently selected clonotypes should have a *lower* generative probability than typical convergently recombined clonotypes. To test this, we estimated the generative probability of the TCR$\beta$'s Complementarity Determining Region 3 (CDR3) amino-acid sequences that were shared between several patients. Since no algorithm exists that can compute this generative probability directly, our method relies on the random generation and translation of massive numbers of TCR nucleotide sequences using a mechanistic statistical model of V(D)J recombination (*Murugan et al., 2012*), as can be easily performed for example using the IGoR software (*Marcou et al., 2017*).

In *Figure 2A* we plot for each clonotype the number of donors sharing that clonotype against its generation probability. Disease-specific TCR$\beta$ variants validated by functional tests in source studies are circled in red. Note that validated disease-specific TCR$\beta$ sequences have a much lower generation probability than the typical sequences shared by the same number of donors. We developed a method of axis transformation (see Materials and methods) to compare the model prediction with data values on the same scale (*Figure 2B*), so that outliers can be easily identified by their distance to identity line. Our method can be used to narrow down the potential candidates for further experimental validation of responsive receptors. Additional information, like the expansion of the identified TCR$\beta$ clonotype in the inflammation site, the presence of the same clonotype in the repertoire of activated or memory T-cells, or absence in a cohort of healthy controls, could provide additional evidence for functional association of identified candidates with a given condition.

Our method also identifies other significant outliers than reported in the source studies (shown in red, and obtained after multiple-test correction – see Materials and methods), which may have three possible origins. First, they may be associated with the condition, but were missed by the source studies. Second, they may be due to other factors shared by the patients, such as features involved in thymic or peripheral selection, or reactivity to other common conditions than CMV (e.g. influenza

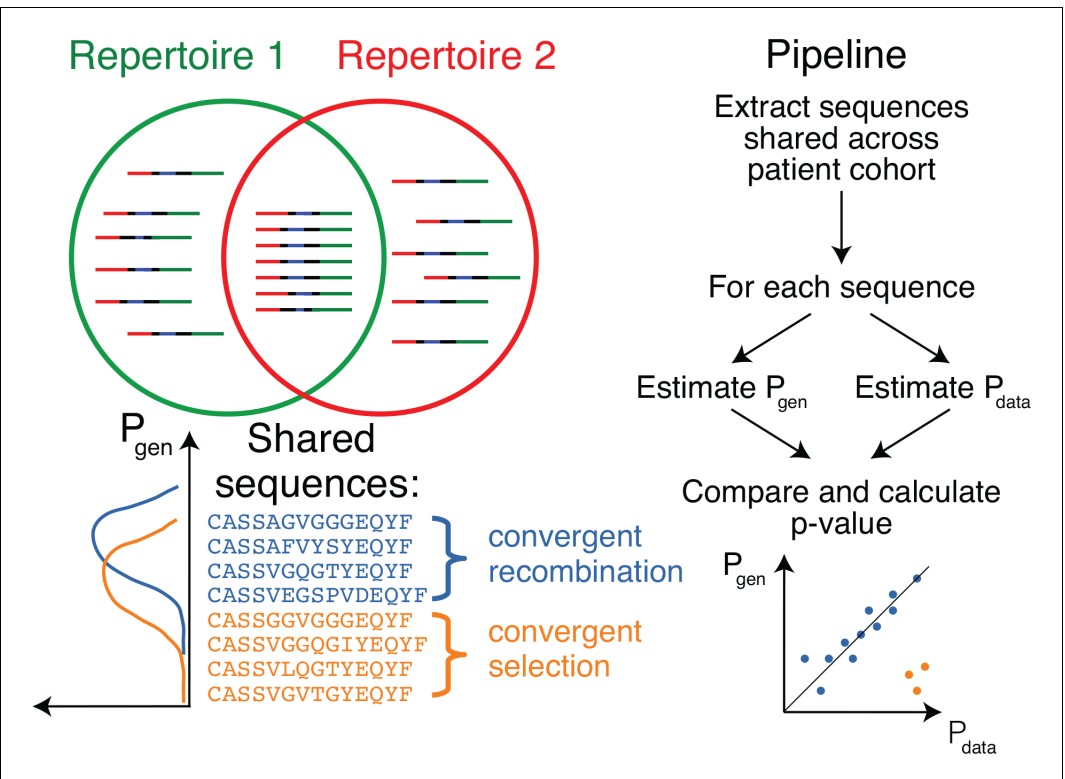

**Figure 1.** Method principle and pipeline. (Top left) Sequence overlap between two TCR or BCR repertoires. (Bottom left) There are two major mechanisms for sequence sharing between two repertoires: convergent recombination and convergent selection. Because convergent recombination favors sequences with high generation probabilities, these two classes of sequences have different distributions of the generative probability, $P_{\mathrm{gen}}(\sigma)$. (Right) We estimate the theoretical $P_{\mathrm{gen}}(\sigma)$ for each sequence $\sigma$ and compare it to $P_{\mathrm{data}}(\sigma)$, which is empirically derived from the sharing pattern of that sequence in the cohort. Comparison of these two values allows us to calculate the analog of a p-value, namely the posterior probability that the sharing pattern is explained by the convergent recombination alone, with no selection for a common antigen.
DOI: https://doi.org/10.7554/eLife.33050.002

infection). Third, they can be the result of intersample contamination. Our approach is able to diagnose the last explanation by estimating the likelihood of sharing at the level of nucleotide sequences (i.e. synonymously), as detailed in the Materials and methods section.

## Discussion

Antigen receptor sequencing currently has little clinical applications. One of the most important ones is diagnostics and tracking of malignant T-cell and B-cell clones in lymphomas, where it allows for directly measuring the abundances of certain clones at different timepoints. Our method allows for a sequence-based theoretical prediction of T-cell abundances at the population level, and for the identification of T-cell clones associated with infectious and autoimmune conditions. Extensive databases of condition-associated clones can provide a means of disease diagnostics and extend the clinical utility of antigen receptor repertoire sequencing technologies.

This method may also be useful in the analysis of known antigen-specific TCR clonotypes. The typical source of such TCR sequences are MHC-multimer positive cells isolated from one or a few donors (*Shugay et al., 2018*; *Tickotsky et al., 2017*). Some of these antigen-specific clonotypes are private, and are hard to find in other patients, providing limited diagnostic value. Our method is able to distinguish these clones from publicly responding clonotypes that are likely to be shared by many patients using only their CDR3 amino acid sequences.

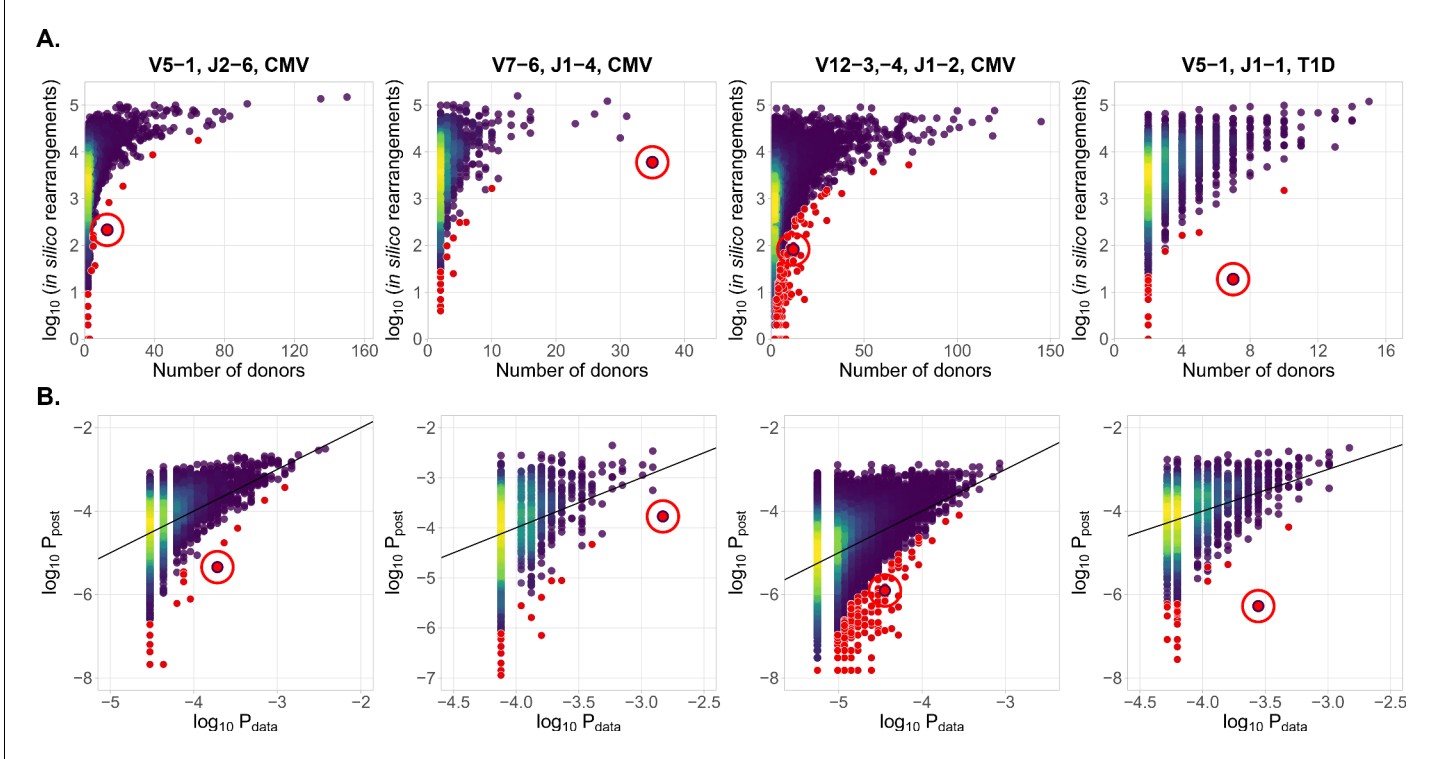

**Figure 2.** Identification of condition-associated clonotypes using generative probability (**A**) CDR3aa of antigen specific clonotypes (red circles) have less generative probability than other clonotypes shared among the same number of donors. The number of in silico rearrangements obtained for each TCR$\beta$ sequence in our simulation (which is proportional to generation probability for each clonotype in a given VJ combination $P_{\text{post}}(\sigma)$), plotted against the number of patients with that TCR$\beta$ clonotype. (**B**) Model prediction of generative probabilities agrees well with data. To directly compare $P_{\text{post}}(\sigma)$ to data, we estimate the empirical probability of occurrence of sequences, $P_{\text{data}}(\sigma)$, from its sharing pattern across donors (see Materials and methods). In A. and B. red dots indicate significant results (adjusted $P<0.01$, Holm's multiple testing correction), while red circles point to the responsive clonotypes identified in the source studies.

DOI: https://doi.org/10.7554/eLife.33050.003

The cohort size necessary for the identification of antigen-specific clonotypes with our method varies (see 'Designing the experiment' subsection in Materials and methods). It depends on the strength and diversity of the response to the given antigen. CMV and other *Herpesviridae* (EBV, HSV), are able to cause a persistent infection, and a large fraction of the TCR repertoire of CMV-positive donors are believed to be specific to them—on average, up to 10% of CD8 +cells are specific to a single CMV epitope in elderly individuals (*Khan et al., 2004*). However, it was shown that in a human acute infection model of yellow fever vaccination, virus-specific T-cell clones are one of the most abundant in the TCR repertoire and occupy up to 12% of the CD8 +T cell repertoire. This response is short-lived and contracts significantly a month after immunization (*Miller et al., 2008*). So the peak of an immune response is the best timepoint to search for antigen-specific TCRs in acute infections using this method. T-cell response to herpesviruses is also not unique in terms of public clonotype involvement—in ankylosing spondylitis (*Faham et al., 2017*), 30–40% of patients share a certain TCR$\beta$ aminoacid sequence, which is more than the fraction of patients sharing CMV-specific clonotypes that we analysed in this study.

Our approach can be used on other hypervariable receptor chains (TCR$\alpha$, BCR heavy and light chains), as well as other species (mice, fish, etc.). Both $\alpha$ and $\beta$ chains contribute to T-cell receptor specificity. Single-cell or paired sequencing technologies (*Zemmour et al., 2018*) could identify partner receptor chains for condition-associated TCR $\alpha$ or $\beta$ chain sequences identified with our approach. Antigenic peptides recognized by complete T-cell receptors could then be recovered in vitro using yeast-display libraries of peptide-MHC (*Gee et al., 2018*). As paired sequencing becomes

more widespread, our method can be extended to the analysis of full paired TCR by applying the exact same analysis using the joint recombination probability of $\alpha\beta$ clonotypes.

Recent advances in computational methods allow us to extract TCR repertoires from existing RNA-Seq data (*Bolotin et al., 2017*; *Brown et al., 2015*). Huge numbers of available RNA-Seq datasets from patients with various conditions can be used for analysis and identification of novel virus, cancer, and self reactive TCR variants using our method. The more immunoglobulin receptors with known specificity are found using this type of association mapping, the more clinically relevant information can be extracted from immunoglobulin repertoire data.

# Materials and methods

## Statistical analysis
### Problem formulation
Our framework is applicable to analyze the outcome of a next generation sequencing experiment probing the immune receptor repertoires of $n$ individuals with a given condition, for example CMV or Type one diabetes. We denote by $M_i$ the number of unique amino acid TCR sequences in patient $i$, $i = 1, \ldots, N$. For a given TCR amino acid sequence $\sigma$, we set $x_i = 1$ to indicate that $\sigma$ is present in patient $i$'s repertoire, and $x_i = 0$ otherwise. For a given shared sequence $\sigma$, we want to know how likely its sharing pattern is under the null hypothesis of convergent recombination, correcting for the donors' different sampling depths. In other words, is $\sigma$ overrepresented in the population of interest? If $\sigma$ is significantly overrepresented, we also want to quantify the size of this effect.

### Overview
Under the null hypothesis, the presence of $\sigma$ in a certain number of donors is explained by independent convergent V(D)J recombination events in each donor. Given the total number of recombination events that led to the sequenced sample of donor $i$, $N_i$, the presence of given amino acid sequence $\sigma$ in donor is Bernoulli distributed with probability

$$p_i = \langle x_i \rangle = \left(1 - P_{post}(\sigma)\right)^{N_i}, \tag{1}$$

$$P_{post}(\sigma) = P_{gen}(\sigma) \times Q, \tag{2}$$

where $P_{\mathrm{post}}(\sigma)$ is the model probability that a recombined product found in a blood sample has sequence $\sigma$ under the null hypothesis. It is formed by the product of $P_{\mathrm{gen}}(\sigma)$, the probability to generate the sequence $\sigma$, estimated using a V(D)J recombination model (see the following *subsection*), and $Q$, a constant correction factor accounting for thymic selection (see *Estimation of the correction factor Q subsection*). The number of independent recombination events $N_i$ leading to the observed unique sequences in a sample $i$ is unknown, because of convergent recombination events within the sample, but it can be estimated from the number of unique sequences $M_i$, using the model distribution $P_{\mathrm{post}}$ (see *Estimation of $N_i$ subsection*).

We also calculate the posterior distribution of $P_{\mathrm{data}}(\sigma)$, corresponding to the empirical counterpart of $P_{\mathrm{post}}(\sigma)$ in the cohort, inferred from the sharing pattern of $\sigma$ across donors. We use information about the presence of $\sigma$ in our donors, $x_1, \ldots, x_n$ and the sequencing depth for each donor, $N_1, \ldots, N_n$ (see *Estimation of $P_{\mathrm{data}}(\sigma)$ subsubsection*), yielding the posterior density: $\rho(P_{\mathrm{data}}|x_1, \ldots, x_N)$.

Finally, we estimate the probability, given the observations, that the true value of $P_{\mathrm{data}}$ is smaller than the theoretical value $P_{\mathrm{post}}$ predicted using V(D)J recombination model, analogous to a p-value and used to identify significant effects:

$$\mathbb{P}(P_{\mathrm{post}} > P_{\mathrm{data}}) = \int_0^{P_{\mathrm{post}}} \rho(P_{\mathrm{data}}|x_1, \ldots, x_n) dP_{\mathrm{data}}. \tag{3}$$

To estimate the effect size $q(\sigma)$ we compare $P_{\mathrm{data}}$ to $P_{\mathrm{post}}$,

$$q(\sigma) = \frac{P_{\mathrm{data}}(\sigma)}{P_{\mathrm{post}}(\sigma)}. \tag{4}$$

## Estimation of $P_{\text{gen}}$, the probability of generation of a TCR CDR3 amino acid sequence

To procedure outlined above requires to calculate $P_{\text{gen}}(\sigma)$, the probability to generate a given CDR3 amino acid sequence. Methods exist to calculate the probability of TCR and BCR nucleotide sequences from a given recombination model (*Murugan et al., 2012*; *Marcou et al., 2017*), but are impractical to calculate the probability of amino acid sequences, because of the large number of codon combinations that can lead to the same amino acid sequence, $\prod_{a=1}^{L} n_{\text{codons}}(\sigma(a))$, where $L$ is the sequence length, and $n_{\text{codons}}(\tau)$ the number of codons coding for amino acid $\tau$. The number is about $1.4 \times 10^7$ for a typical CDR3 length of 15 amino acid.

Instead, we estimated $P_{\text{gen}}(\sigma)$ using a simple Monte-Carlo approach. We randomly generated a massive number ($N_{\text{sim}} = 2 \times 10^9$) of recombination scenarios according to the validated recombination model (*Murugan et al., 2012*):

$$
\begin{aligned}
P_{\text{rearr}}^{\beta} &= P(V)P(D,J)P(\text{del}V|V)P(\text{ins}VD) \\
&\times P(\text{del}Dl, \text{del}Dr|D)P(\text{ins}DJ)P(\text{del}J|J).
\end{aligned}
\tag{5}
$$

The resulting sequences were translated, truncated to only keep the CDR3, and counted. $P_{\text{gen}}(\sigma)$ was approximated by the fraction of events thus generated that led to sequence $\sigma$. This approximation becomes more accurate as $N_{\text{sim}}$ increases, with an error on $P_{\text{gen}}(\sigma)$ scaling as $(P_{\text{gen}}(\sigma)/N_{\text{sim}})^{1/2}$.

## Estimation of the correction factor $Q$

Not all generated sequences pass selection in the thymus. $P_{\text{gen}}$ systematically underestimates the frequency of recombination event that eventually make it into the observed repertoire. To correct for this effect, we estimate a correction factor $Q$, as was suggested in (*Elhanati et al., 2014*):

$$
P_{\text{post}}(\sigma) = P_{\text{gen}}(\sigma) \times Q.
\tag{6}
$$

Contrary to (*Elhanati et al., 2014*), which learned a sequence-specific factor for each individual, here we assume that all observed sequences passed thymic selection. $Q$ is a normalization factor accounting for the fact that just a fraction $Q^{-1}$ of sequences pass thymic selection. This factor is determined for each VJ-combination as an offset when plotting $\log P_{\text{gen}}$ against $\log P_{\text{data}}^{*}$ (see the following *subsection* for definition of $P_{\text{data}}^{*}$), using least squares fitting.

## Estimation of $P_{\text{data}}(\sigma)$, the probability of sequence occurrence in data

The variable $x_i$ indicates the presence or absence of a given TCR amino acid sequence $\sigma$ in the $i$th dataset with $N_i$ recombination events per donor. We want to estimate $P_{\text{data}}(\sigma)$, which is a fraction of recombination events leading to $\sigma$ in the population of interest. According to Bayes' theorem, for a given $\sigma$, the probability density function of $P_{\text{data}}$ reads:

$$
\rho(P_{\text{data}}|x_1, \ldots, x_n) = \frac{\mathbb{P}(x_1, \ldots, x_n|P_{\text{data}})\rho_{\text{prior}}(P_{\text{data}})}{\int_0^1 \mathbb{P}(x_1, \ldots, x_n|P_{\text{data}})\rho_{\text{prior}}(P_{\text{data}})\,dP_{\text{data}}}.
\tag{7}
$$

The likelihood is given by a product of Bernouilli probabilities:

$$
\mathbb{P}(x_1, \ldots, x_n|P_{\text{data}}) = \prod_{i=1}^{N} \left[1 - (1 - P_{\text{data}})^{N_i}\right]^{x_i} \left[(1 - P_{\text{data}})^{N_i}\right]^{1-x_i},
\tag{8}
$$

and a flat prior $\rho_{\text{prior}}(P_{\text{data}}) = \text{const}$ is used.

We estimate $P_{\text{data}}^{*}$ (shown in *Figure 2B*) as the maximum of the posterior distribution:

$$
P_{\text{data}}^{*} = \arg\max_{P_{\text{data}}} \rho(P_{\text{data}}|x_1, \ldots, x_n).
\tag{9}
$$

## Estimation of $N_i$, the number of recombination events

The total number $N_i$ of recombination events in $i$th dataset is unknown, but we can count the number of unique CD3 acid sequences $M_i$ observed in the sequencing experiment. For a typical TRB experiment, convergent recombination is relatively rare and one could use $N_i \approx M_i$ as an

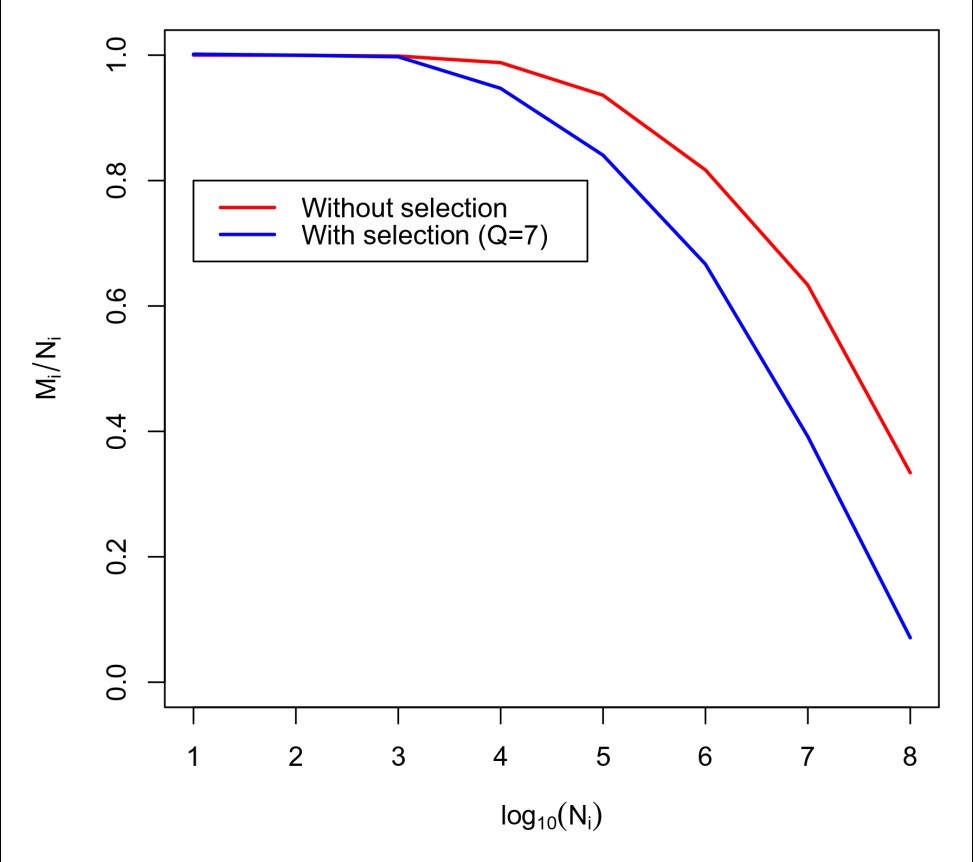

**Figure 3.** Calibration curve for TRBV5-1 TRBJ2-6 combination. Here we plot the fraction of unique amino acid sequences to recombination events against the logarithm of the number of recombination events. The blue line corresponds to the theoretical solution with selection, the red line corresponds to the theoretical solution without selection.

DOI: https://doi.org/10.7554/eLife.33050.004

approximation. However, for less diverse loci (e.g TRA), or for much higher sequencing depths, one should correct for convergent recombination, as the the observed number of unique aminoacid sequences could be much lower than the actual number of corresponding recombination events.

The average number of unique sequences resulting from $N_i$ recombination events is, in theory:

$$\langle M_i \rangle = \sum_{\sigma \in T} (1 - P_{\text{post}}(\sigma))^{N_i}. \tag{10}$$

where $T$ is the set of sequences that can pass thymic selection. To estimate that number, we generate a very large number $N_{\text{sim}}$ of recombinations, leading to $N_{\text{uni}}$ unique CDR3 amino acid sequences for which $P_{\text{gen}}$ is estimated as explained above. We take $T$ to be a random subset of unique sequences, $T \subset \{\sigma_1, \ldots, \sigma_{N_{\text{uni}}}\}$, of size $|T| = N_{\text{uni}}/Q$, and we apply *Equation 8*.

Using this equation we plot the calibration curve for the TRBV5-1 TRBJ2-6 VJ datasets in *Figure 3*. For comparison the case of no thymic selection ($Q = 1$) is shown in red. The inversion of this curve yields $N_i$ as a function of $M_i$.

## Pipeline description

In this section we describe how to apply our algorithm to real data. All the code and data necessary to reproduce our analysis is available online on github (https://github.com/pogorely/vdjRec/; copy archived at https://github.com/elifesciences-publications/vdjRec/).

We start with annotated TCR datasets (CDR3 amino acid sequence, V-segment, J-segment), one per donor. Such datasets are produced by MiXCR (*Bolotin et al., 2015*), immunoseq (http://www.

adaptivebiotech.com/immunoseq) and most other software for NGS repertoire data preprocessing. Data we used was in immunoseq format, publicly available from https://clients.adaptivebiotech.com/immuneaccess database.

We proceed as follows:

1. Split datasets by VJ combinations. The resulting datasets correspond to lists of unique CDR3 amino acid sequences for each donor and VJ combination. All following steps should be done independently for each VJ combination.
2. (Optional). Filter out sequences present in only one donor to speed up the downstream analysis.
3. Generate a large amount of simulated nucleotide TCR sequences for a given VJ combination. Extract and translate their CDR3, and count how many times each sequence appears in the simulated set (restricting to sequences actually observed in donors for better efficiency). The resulting number divided by the total number of simulated sequences is an estimate of $P_{\text{gen}}$.
4. Estimate $P_{\text{data}}^*$ for each sequence in the dataset, see *Estimation of $P_{\text{data}}(\sigma)$ subsection*.
5. Using $P_{\text{data}}^*$ and $P_{\text{gen}}$, estimate for each VJ combination the normalization $Q$ by minimizing $\sum_{j=1}^{n}(\log P_{\text{data}}^*(\sigma_j) - \log P_{\text{gen}}(\sigma_j) - \log Q)^2$, see *Estimation of the correction factor $Q$ subsection*, where $\sigma_j, j = 1, \ldots, n$ are the shared sequences.
6. Calculate $P_{\text{post}} = Q \times P_{\text{gen}}$. Calculate the p-value (*Equation 1*) and effect size (*Equation 2*).

## Usage example

### Data sources

Data from (*Emerson et al., 2017*) and (*Seay et al., 2016*) is publicly available from the immuneaccess database: https://clients.adaptivebiotech.com/immuneaccess. For our analysis, we only considered VJ combinations for which the authors identified condition-associated clonotypes with MHC-multimer proved specificity. CDR3 aminoacid sequences and V and J segment of these TCR clonotypes are given in *Table 1*.

### Analysis results

We applied our pipeline to identify CMV-specific and self-specific TCR sequences listed in *Table 1*. For our analysis we used only case cohorts, without controls. For each dataset we followed our pipeline described in *Pipeline description subsection*. We found that sequences reported in the source studies as being both significantly enriched in the patient cohort, and antigen-specific according to MHC-multimers, were the most significant in 3 out of 4 datasets (See *Table 2*). In the remaining TRBV12 dataset, the sequence of interest was the top 40 most significant out of $27,699$ sequences present in at least two CMV-positive donors.

## Identifying contaminations

Intersample contamination may complicate high-throughput sequencing data analysis in many ways. It could occur both during library preparation or the sequencing process itself (*Sinha et al., 2017*). Contaminations have the same nucleotide and amino acid sequence in all datasets, and so our method identifies them as outliers, because their sharing cannot be explained by a high recombination probability.

Our method provides a tool to diagnose contamination. Given an amino-acid sequence present in many donors, we measure its theoretical nucleotide diversity using the same simulation approach we used to calculate the generative probability $P_{\text{gen}}$ of the amino acid sequence (see *Estimation of*

**Table 1.** Published antigen-specific clonotypes used to test the algorithm.

| CDR3aa | V-segment | J-segment | Antigen source | Ref. |
|---|---|---|---|---|
| CASSLAPGATNEKLFF | TRBV07-06 | TRBJ1-4 | CMV | (*Emerson et al., 2017*) |
| CASSPGQEAGANVLTF | TRBV05-01 | TRBJ2-6 | CMV | (*Emerson et al., 2017*) |
| CASASANYGYTF | TRBV12-3,−4 | TRBJ1-2 | CMV | (*Emerson et al., 2017*) |
| CASSLVGGPSSEAFF | TRBV05-01 | TRBJ1-1 | self | (*Seay et al., 2016*; *Gebe et al., 2009*) |

DOI: https://doi.org/10.7554/eLife.33050.005

**Table 2.** Output of the algorithm for sequences from *Table 1*.

| CDR3aa | V | J | Ag.source . | p-value rank | p-value | Effect size |
|---|---|---|---|---|---|---|
| CASSLAPGATNEKLFF | 07–06 | 1–4 | CMV | 1/1637 | $1.2 \times 10^{-17}$ | 8.8 |
| CASSPGQEAGANVLTF | 5–01 | 2–6 | CMV | 1/5549 | $1.8 \times 10^{-17}$ | 42.3 |
| CASASANYGYTF | 12–3,−4 | 1–2 | CMV | 40/27669 | $2.5 \times 10^{-14}$ | 28.8 |
| CASSLVGGPSSEAFF | 5–01 | 1–1 | self | 1/2646 | $9.5 \times 10^{-19}$ | 524 |

DOI: https://doi.org/10.7554/eLife.33050.006

$P_{\text{gen}}$ subsection). If the diversity of the simulated nucleotide sequences is much larger than observed in the data, it is a sign of contamination.

We applied this approach to the CDR3 sequence CASSLVGGPSSEAFF associated to Type one diabetes, and found 19 recombination events consistent with that amino acid sequence out of our simulated dataset. We found 18 different nucleotide variants out of the 19 total possible. In contrast, in the data this clononotype had the same nucleotide variant in all of the eight donors in which it was present. That variant was absent from the simulated set. A one-sided Fisher exact test gives a $p < 10^{-6}$ probability of this happening by chance, indicating contamination as a likely source of sharing.

## Designing the experiment

Our approach also allows us to obtain important estimates for experiment design. A number of variables affect detection of an antigen-specific clone using our approach: the abundance of the clone in the general population (represented by $P_{\text{data}}$ in our approach), the cohort size, the sequencing depth $N_i$ in each donor in the cohort, and also the effect size. Fixing any two of these variables results in a constraint between the other two and the affects the probability to detect an antigen-specific clonotype, which translates into the statistical power of the method. As an example of such an analysis, we fix the cohort size at 10, 30 or 100 donors (see *Figure 4A–C* respectively) and the sequencing depth at $N_i = 1000$ unique clones sequenced per repertoire for a given VJ-combination in each donor in the cohort. We ask how frequently a disease specific clone with $\tilde{P}_{\text{data}}$ abundance in the population and effect size $q = \tilde{P}_{\text{data}}/P_{\text{post}}$ is detected with our method. To address this question for each value $\tilde{P}_{\text{data}}$ we perform a simulation: we simulate $x_1, x_2, \ldots, x_n$ Bernoulli variables, each with a $p_i = 1 - e^{-N_i \tilde{P}_{\text{data}}}$ success probability. For a given value of $\tilde{P}_{\text{data}}$ and $q$ there is a single value of $P_{\text{post}} = \tilde{P}_{\text{data}}/q$. Then we calculate

$$\mathbb{P}(P_{\text{post}} > P_{\text{data}}) = \int_0^{P_{\text{post}}} \rho(P_{\text{data}}|x_1, \ldots, x_n) dP_{\text{data}}, \tag{11}$$

where $\rho(P_{\text{data}}|x_1, \ldots, x_n)$ is the posterior density, and check if $\mathbb{P}(P_{\text{post}} > P_{\text{data}})$ is below a significance threshold of 0.0001. Such a low significant threshold in this example is chosen to take into account the multiple testing correction: we assume that about 1000 shared clones would be tested in a such analysis and p¡0.01 after multiple testing is chosen as the significance threshold in this study, which gives p¡0.0001 before the Bonferroni multiple testing correction. Then we plot the number of simulations in which a significant result was obtained for given effect size $q$ and $\tilde{P}_{\text{data}}$ for the clone of interest and the fraction of donors with this sequence in the simulated cohort (see *Figure 4E*, blue curve). Unsurprisingly, the effect size plays a role in the probability to detect an antigen specific clone, and the detection is not possible at all if the clone is not shared between several donors in the cohort (in our example this happens for $\tilde{P}_{\text{data}} < 10^{-5}$) irrespective to the effect size. Larger cohort sizes can help to resolve clones with lower abundances, but sequencing depth also has a strong effect on the power of the approach. In *Figure 4D and E* we show simulation results for a fixed $q = 10$ and different sequencing depths $N_i$ of 100, 1000 or 10000 clones per donor in a given VJ combination. Interestingly, a large sequencing depth (black curve) can lead to a situation when an abundant and frequently generated clone will not be detected by the algorithm, because it will be found in all donors in the cohort. An additional test that checks the predictions by lowering the sequencing depth in silico by downsampling can solve this problem.

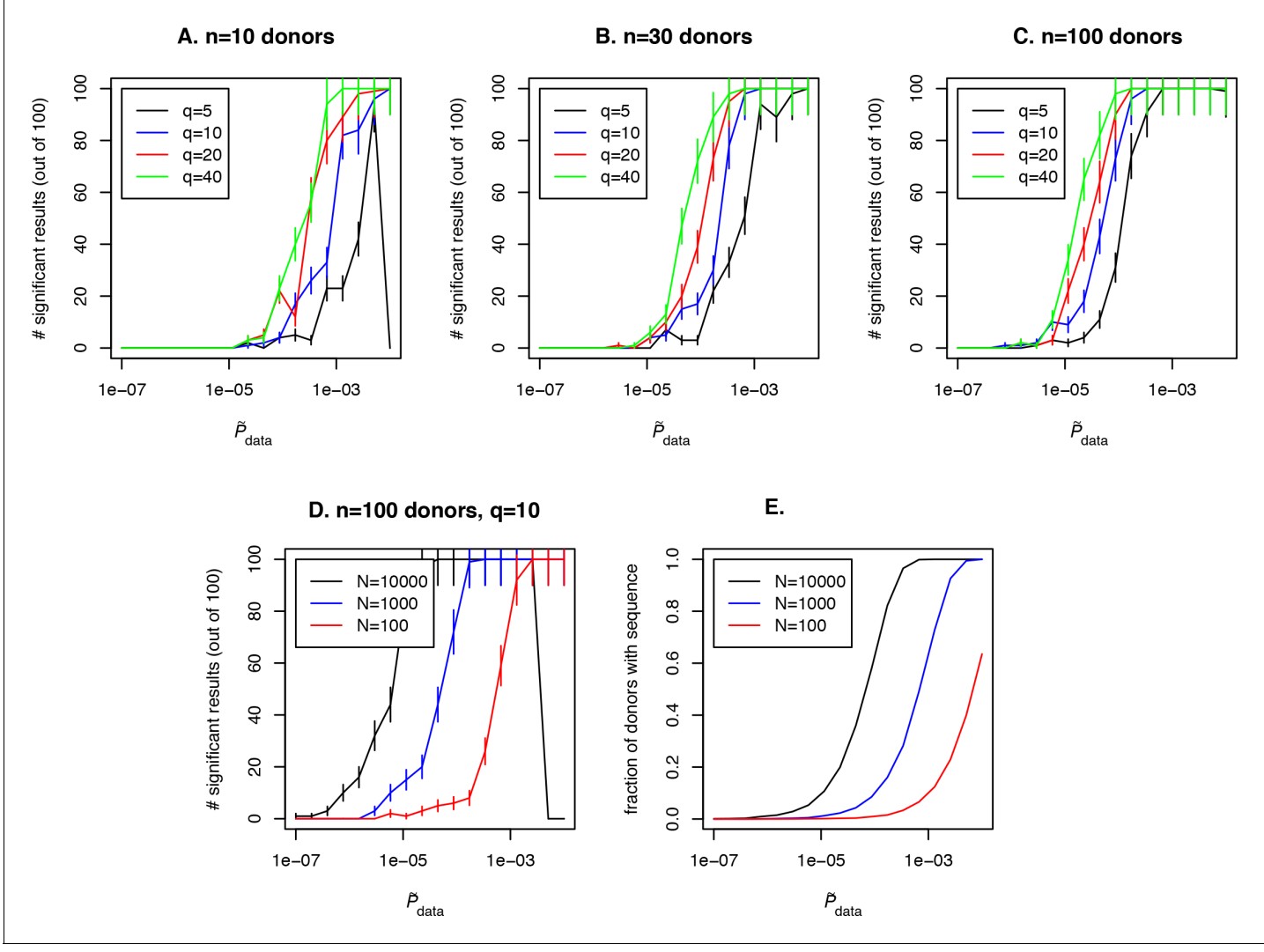

**Figure 4.** Simulation of the method performance with different cohort sizes, sequencing depths, effect sizes and target clone abundances in population. In panels (**A. B. C**) we plot the number of simulations (out of 100) where a clone with a given effect size $q$ (line color, see legend) and $\tilde{P}_{\mathrm{data}}$ (x-axis) is found to be significant using our approach, for cohort sizes of 10, 30 and 100 donors respectively. Larger cohort sizes and effect sizes make it possible to resolve clonotypes with lower abundance in the population. In panel (**D**) we show the effect of sequencing depth for fixed $q = 10$: larger numbers of clonotypes sequenced per donor allow us to resolve less frequent clones, since a clone of a given $\tilde{P}_{\mathrm{data}}$ is detected in a larger fraction of donors (panel **E**).

DOI: https://doi.org/10.7554/eLife.33050.007

Another complicated question is how $P_{\mathrm{data}}$ is related to the number of clones and the fraction of the repertoire involved in the response to the infection in a given donor. If the same antigen-specific clone is present in every donor, $P_{\mathrm{data}}$ is close to the average abundance of this clone in the repertoire. However one can imagine an opposite situation where the response is so diverse and private that different clones respond to a given antigen in each donor. It was previously shown that the diversity and publicness of responding T-cell clonotypes varies a lot across antigens (**Dash et al., 2017**). Our approach is restricted to the identification of *public* antigen-specific clonotypes, which may not exist for all antigens.

## Acknowledgements

This work was supported by Russian Science Foundation grant 15-15-00178, and partially supported by European Research Council Consolidator Grant 724208.

## Additional information

### Competing interests
Aleksandra M Walczak: Reviewing editor, *eLife*. The other authors declare that no competing interests exist.

### Funding

| Funder | Grant reference number | Author |
|---|---|---|
| Russian Science Foundation | 15-15-00178 | Dmitriy M Chudakov<br>Ilgar Z Mamedov<br>Yuri B Lebedev |
| European Research Council | 724208 | Aleksandra M Walczak |

The funders had no role in study design, data collection and interpretation, or the decision to submit the work for publication.

### Author contributions
Mikhail V Pogorelyy, Conceptualization, Data curation, Software, Formal analysis, Validation, Investigation, Visualization, Methodology, Writing—original draft, Writing—review and editing; Anastasia A Minervina, Data curation, Software, Investigation; Dmitriy M Chudakov, Ilgar Z Mamedov, Yuri B Lebedev, Conceptualization, Resources, Supervision, Funding acquisition, Investigation, Methodology, Project administration, Writing—review and editing; Thierry Mora, Conceptualization, Resources, Formal analysis, Supervision, Funding acquisition, Investigation, Methodology, Writing—original draft, Project administration, Writing—review and editing; Aleksandra M Walczak, Conceptualization, Resources, Supervision, Funding acquisition, Investigation, Methodology, Writing—original draft, Project administration, Writing—review and editing

### Author ORCIDs
Mikhail V Pogorelyy (iD) http://orcid.org/0000-0003-0773-1204
Dmitriy M Chudakov (iD) https://orcid.org/0000-0003-0430-790X
Thierry Mora (iD) https://orcid.org/0000-0002-5456-9361
Aleksandra M Walczak (iD) http://orcid.org/0000-0002-2686-5702

### Decision letter and Author response
Decision letter https://doi.org/10.7554/eLife.33050.014
Author response https://doi.org/10.7554/eLife.33050.015

## Additional files

### Supplementary files
• Transparent reporting form
DOI: https://doi.org/10.7554/eLife.33050.008

### Major datasets
The following previously published datasets were used:

| Author(s) | Year | Dataset title | Dataset URL | Database, license, and accessibility information |
|---|---|---|---|---|
| Emerson RO, De-Witt WS, Vignali M, Gravley J, Hu JK, Osborne EJ, Desmarais C, Klinger M, Carlson CS, Hansen JA, Rieder M, Robins HS | 2017 | Immunosequencing identifies signatures of cytomegalovirus exposure history and HLA-mediated effects on the T cell repertoire | http://dx.doi.org/10.21417/B7001Z | Publicly available at ImmuneAccess. |

| | | | | |
|---|---|---|---|---|
| Seay HR, Yusko E, Rothweiler SJ, Zhang L, Posgai AL, Campbell-Thompson M, Vignali M, Emerson RO, Kaddis JS, Ko D, Nakayama M, Smith MJ, Cambier JC, Pugliese A, Atkinson MA, Robins HS, Brusko TM | 2016 | Tissue distribution and clonal diversity of the T and B cell repertoire in type 1 diabetes | http://doi.org/10.21417/B73S3K | Publicly available at ImmuneAccess |

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
