## [Decision Letter]

Thank you for submitting your article "Method for identification of condition-associated public antigen receptor sequences" for consideration by *eLife*. Your article has been reviewed by two peer reviewers, and the evaluation has been overseen by a Reviewing Editor and Arup Chakraborty as the Senior Editor. The reviewers have opted to remain anonymous.

The reviewers have discussed the reviews with one another and the Reviewing Editor/Senior Editor has drafted this decision to help you prepare a revised submission.

Your manuscript describes a computational pipeline that identifies particular TCR sequences that are over represented in groups of individuals. In effect, it identifies the outlier TCR sequences populations from a computationally determined normal distribution. You go on to show that, based on TCR deep sequencing datasets, this method can retrospectively identify TCRβ chains that are over represented following CMV infections, and TCRβ sequences that have reactivity to defined autoantigens in type 1 diabetes. The manuscript is theoretically interesting, useful, timely, and clearly written. However, we have some significant concerns about the general applicability of this tool and its value for prospective, rather than retrospective, studies. These concerns are listed below and need to be addressed in a revision.

1) Methods to identify outlier populations are understood and statistical approaches to achieve this goal are available. Indeed, a related approach to this question is the basis for lymphoid tumor diagnostic methodologies being pursued by several companies. In what ways is your tool different/better compared to the available methods?

2) Your work is based on small cohorts of patients. HCMV can be responsible for a disproportionate number of TCR clonotypes and can have a stable profile over many years, even by the standards of other chronic infections. So, we wonder whether an attractive feature of this tool, which is its ability to glean information from data on small cohorts will carry over beyond the HCMV setting. Would one need a larger cohort for most disease applications? Is there a theoretical way to estimate what part of the repertoire (in terms of unique clones and their frequencies) has to be devoted to an infectious disease in order for it to be resolved in a given cohort size (maybe a scaling rule)?

3) A general concern is whether this tool will have broad impact. You used TCR sequence data sets of HLA-matched patients, which had already been shown to have T cell expansion to a defined peptide/HLA combination. The approach works if one has HLA-matched subjects, because one would assume in general that different HLA alleles will present a different spectrum of peptides, and that TCR Vgene – HLA interactions impact ligand specificity. Further, a significant element to the approach only works retrospectively; without prior knowledge of the TCR specificity one does not know whether the identified outlier populations are disease-relevant. For example, in Figure 2, there are many outlier sequences (red), which may be noise, may be disease relevant or may be consequences of HLA-biased selection that was not accounted for by the authors computationally-defined normal TCRb distribution. Can the method be useful for prospective studies, or learn new biology in retrospective studies? Perhaps one way to address this question is the following:

Is there any other viral infection for which comparable data is available? For instance, it would be nice if data on viruses like HSV or EBV were available along with data related to acute infection or vaccination (where a method such as this, if applicable, would be incredibly valuable).

[Editors' note: further revisions were requested prior to acceptance, as described below.]

Thank you for resubmitting your work entitled "Method for identification of condition-associated public antigen receptor sequences" for further consideration at *eLife*. Your revised article has been favorably evaluated by Arup Chakraborty (Senior editor), a Reviewing editor, and 1 reviewer.

You present an important statistical tool for analyzing TCR sequence diversity. We will soon get to a point where entire human TCR repertoires will be sequenced and then the question is how do we understand this data to assess immune health – your method is an advance that allows us to take steps toward this goal, and is therefore a valuable resource/tool. However, you often confuse TCR with clonotype in the writing of your paper. A TCR β chain sequence is not a TCR clonotype. Throughout the manuscript, you describe what you are analyzing as TCR clonotypes when you are actually referring to independent TCR β sequences. An independent TCR α chain or a TCR β chain sequence is NOT a receptor. Once this point is consistently clarified in the manuscript, your paper will be accepted.

Reviewer #1:

Re-review:

The major concern I have with the method, writing and the inability of this platform to provide a framework for prospective studies is the author's miss-statements (I assume they know better) that a TCR β chain sequence is not a TCR clonotype. Throughout the manuscript, they authors describe what they are analyzing as TCR clonotypes when they are actually referring to independent TCR β sequences. An independent TCRa chain or a TCRb chain sequence is NOT a receptor. The CDR3beta sequence does not equal or even mark a particular T cell clone.

Problem statement that this creates:

Example in the Introduction: the authors argue that there are "Several mechanisms leading to the repertoire overlap. The first mechanism is convergent recombination[…]" This first statement or option is a product of sequencing just the TCRb chain, i.e., there are common rearrangements that occur do to V-J or V-D-J rearrangements that have no or few N-region additions. The second and third "options" are indeed TCRa + TCRb clonotypes, comparing the first possibility to 2 and 3 is comparing apples to oranges.

Due to biases in V(D)J recombination process, the probability of generation of some receptors is very high, making them appear in almost every individual multiple times and repeatedly sampled in repertoire profiling experiments Britanova et al., (2014). This sharing does not result from a common specificity or function of the shared clonotypes and may in fact correspond to cells from the naive compartment in both donors Quigley et al., (2010), or from functionally distinct subsets such as CD4 and CD8 T-cells.

The second possible reason for TCR sequence sharing is specific to identical twins, who may share T cell clones as a consequence of cord blood exchange in utero via a shared placenta Pogorelyy et al., (2017). The third and most interesting mechanism for sharing receptor sequences is convergent selection, in response to a common antigen.

- The terminology makes these types of statements nonsensical. Identical TCR clonotypes will absolutely have the identical antigen-specificity.

Issues that miss-representing TCRb sequences for TCR clonotypes in the general approach being promoted:

The incorrect use of TCR clonotype leads to some particularly difficult to imagine scenarios. For examples, in the reviewer response and within the manuscript, "our method can be used to narrow down the potential candidates for further experimental validation of responsive receptors." (i.e., identify the antigen being recognized). Their idea regarding this approach is to use: "Functional tests (like cultivation with peptides, or cytokine secretion assays) are the ultimate way to confirm specificity of these predicted clonotypes."

Within the cover letter: "We also are careful to note that this method should be used to identify antigen-specific candidates that need to be further verified by other methods. Nevertheless, we believe that identifying candidates for these experiments is extremely useful."

- However, the sequencing and analyses method of the manuscript does not identify the V α chain, with only half of the receptor there is nothing to study further. Thus, there is no method provided that one could identify interesting receptors for prospective studies.

In summary, this is a method to identify outlier populations, for which retrospective data can be superimposed to create a snapshot of an immune response that has already been described using for example pMHC tetramers or other methods.

---

## [Author Response]

Your manuscript describes a computational pipeline that identifies particular TCR sequences that are over represented in groups of individuals. In effect, it identifies the outlier TCR sequences populations from a computationally determined normal distribution. You go on to show that, based on TCR deep sequencing datasets, this method can retrospectively identify TCRβ chains that are over represented following CMV infections, and TCRβ sequences that have reactivity to defined autoantigens in type 1 diabetes. The manuscript is theoretically interesting, useful, timely, and clearly written. However, we have some significant concerns about the general applicability of this tool and its value for prospective, rather than retrospective, studies. These concerns are listed below and need to be addressed in a revision.1) Methods to identify outlier populations are understood and statistical approaches to achieve this goal are available. Indeed, a related approach to this question is the basis for lymphoid tumor diagnostic methodologies being pursued by several companies. In what ways is your tool different/better compared to the available methods?

Current approaches for lymphoid tumor and minimal residual disease (MRD) diagnostics rely on longitudinal tracking of certain malignant rearrangements before and after treatment. These technologies use the abundance of the malignant clones in the organism before and after the treatment, which is measured directly with NGS, qPCR or flow cytometry. The sequences of the malignant clones are not linked to a known function. In our method we use a sequence-based theoretical estimate of a TCR clone’s abundance in the population to identify candidates for functional receptors that are convergently selected in different patients in response to a specific disease, which is a different task than the direct longitudinal tracking of clonotypes. However, there is a possible lesson to be learned from the overlap between the two methods: tracking malignant clones that have a high recombination probability may not be reliable because one could find an independently recombined healthy clone with the same sequence and interpret it as malignant. Conversely, the TCR recombination probability may be used to identify malignant TCR clones not suitable for longitudinal tracking due to their high probability to be recombined multiple times (Nazarov et al., 2016). If such clonotypes are found after chemotherapy in lymphoma patients, they could be surviving malignant clonotypes as usually assumed, but alternatively they could be different healthy clones with exactly the same independently recombined TCR sequence. In this case it is better to use different MRD markers and the antigen receptor generation probability could help to identify this situation.

We added the following text to clarify the features of our method (Discussion section):

“Antigen receptor sequencing currently has little clinical applications. One of the most important ones is diagnostics and tracking of malignant T-cell and B-cell clones in lymphomas, where it allows for directly measuring the abundances of certain clones at different timepoints. Our method allows for a sequence-based theoretical prediction of T-cell abundances at the population level, and for the identification of T-cell clones associated with infectious and autoimmune conditions. Extensive databases of condition-associated clones can provide a means of disease diagnostics and extend the clinical utility of antigen receptor repertoire sequencing technologies.”

2) Your work is based on small cohorts of patients. HCMV can be responsible for a disproportionate number of TCR clonotypes and can have a stable profile over many years, even by the standards of other chronic infections. So, we wonder whether an attractive feature of this tool, which is its ability to glean information from data on small cohorts will carry over beyond the HCMV setting. Would one need a larger cohort for most disease applications?

Data from published studies suggests that CMV is not a unique condition both in terms of the response magnitude and the number of donors sharing disease specific clones. Faham et al., have shown that 30–40 percent of donors with ankylosing spondylitis share a certain aminoacid TCR sequence, which is absent in healthy controls. This sharing number is larger than the fraction of donors sharing a CMV-specific clone in our current study. We would have liked to analyze different datasets, but we could not access other datasets with suitable cohort sizes and sequencing depth (see below for a discussion of this point).

One can also extend this method to acute infections. It is known that CMV-related clonotypes can occupy a large fraction of the repertoire for a long time [Khan et al., 2004]. However, in an acute viral infection model of yellow fever immunization it was shown that YF-reactive clones that are undetected in the repertoire before immunization can occupy up to 10 percent of the repertoire at the peak of the response (Miller et al., 2008, DeWitt et al., 2015). The response to acute infections is much less stable and strongly decreases on short timescales, but one may overcome this limitation by collecting samples at the peak of the immune response to an infection.

We added the following sentences into the text to clarify this point (Discussion section):

“The cohort size necessary for the identification of antigen-specific clonotypes with our method varies (see “Designing the experiment” subsection in Methods). It depends on the strength and diversity of the response to the given antigen. […] T-cell response to herpesviruses is also not unique in terms of public clonotype involvement – in ankylosing spondylitis (Faham et al., 2017), 30–40 percent of patients share a certain TCR aminoacid sequence, which is more than the fraction of patients sharing CMV-specific clonotypes that we analysed in this study.”

Is there a theoretical way to estimate what part of the repertoire (in terms of unique clones and their frequencies) has to be devoted to an infectious disease in order for it to be resolved in a given cohort size (maybe a scaling rule)?

This is a very interesting question, which we now address with a new section and discussion. In short, the answer depends on both the cohort size and the sequencing depth. To simplify the experiment design, we added an additional pipeline into the software, which is able to tell if a clone of given abundance in the population of patients would be resolved in a given cohort size.

We added an additional subsection in the Materials and methods section (“Designing the experiment”) and a new figure (see Figure 4), demonstrating examples of such analysis.

3) A general concern is whether this tool will have broad impact. You used TCR sequence data sets of HLA-matched patients, which had already been shown to have T cell expansion to a defined peptide/HLA combination. The approach works if one has HLA-matched subjects, because one would assume in general that different HLA alleles will present a different spectrum of peptides, and that TCR Vgene – HLA interactions impact ligand specificity. Further, a significant element to the approach only works retrospectively; without prior knowledge of the TCR specificity one does not know whether the identified outlier populations are disease-relevant. For example, in Figure 2, there are many outlier sequences (red), which may be noise, may be disease relevant or may be consequences of HLA-biased selection that was not accounted for by the authors computationally-defined normal TCRb distribution. Can the method be useful for prospective studies, or learn new biology in retrospective studies?

Our study may seem retrospective, because we chose examples where the right answer (public clonotypes with proved specificity) is known. This was necessary to validate the output of the algorithm. In 3 experiments out of 4 the correct (meaning validated in the source study) clonotype was the most significant signal in our analysis. We also are careful to note that this method should be used to identify antigen-specific candidates that need to be further verified by other methods. Nevertheless, we believe that identifying candidates for these experiments is extremely useful.

Functional tests (like cultivation with peptides, or cytokine secretion assays) are the ultimate way to confirm specificity of these predicted clonotypes. Clonal expansion of identified clonotypes at the inflammation site, presence in the repertoire of activated/memory T-cells, or absence in the control cohort repertoire, may provide additional evidence for condition-association of a given candidate clone.

We added the following sentence into the main text (Results section):

“Additional information, like the expansion of the identified clone in the inflammation site, the presence of the same clonotype in the repertoire of activated or memory T-cells, or absence in a cohort of healthy controls, could provide additional evidence for functional association of identified candidates with a given condition.”

Additionally, there is an interest in purely retrospective studies. In retrospective studies (where the specificity of TCR clones is already defined using, for example, MHC-multimers) it may be useful to predict the abundance of the known antigen-specific clones in the general population. For instance, if a researcher isolated and sequenced antigen-specific T-cells from a patient, some of these T-cell clones could be private and may not be found in other people, while some of them may be public, and easily found in many more yet unstudied patients. Such public clones are much more valuable for diagnostics, and they can be identified using only their TCR aminoacid sequence in such retrospective studies.

We added the following sentences to the main text to clarify this point (Discussion section):

“This method may also be useful in the analysis of known antigen-specific TCR clonotypes. The typical source of such TCR sequences are MHC-multimer positive cells isolated from one or a few donors (Shugay et al., 2017., Tickotsky et al., 2017). Some of these antigen-specific clonotypes are private, and are hard to find in other patients, providing limited diagnostic value. Our method is able to distinguish these clones from publicly responding clonotypes that are likely to be shared by many patients using only their CDR3 amino acid sequence.”

Perhaps one way to address this question is the following:Is there any other viral infection for which comparable data is available? For instance, it would be nice if data on viruses like HSV or EBV were available along with data related to acute infection or vaccination (where a method such as this, if applicable, would be incredibly valuable).

To our knowledge, only one study on response to acute infection is available: DeWitt et al., 2015. We have tried to analyze this dataset but unfortunately, only 9 donors are available in this study, which is not enough to get significant results with our method. The authors also do not report sequences of significantly expanded clonotypes, so it is not possible to validate the output of the algorithm.

As a rule of thumb, assuming currently used typical sequencing depth, our method is applicable for identification of shared TCR sequences in vaccination and acute infection data, but cohorts of at least 20–30 donors are required for such analysis.

[Editors' note: further revisions were requested prior to acceptance, as described below.]

You present an important statistical tool for analyzing TCR sequence diversity. We will soon get to a point where entire human TCR repertoires will be sequenced and then the question is how do we understand this data to assess immune health – your method is an advance that allows us to take steps toward this goal, and is therefore a valuable resource/tool. However, you often confuse TCR with clonotype in the writing of your paper. A TCR β chain sequence is not a TCR clonotype. Throughout the manuscript, you describe what you are analyzing as TCR clonotypes when you are actually referring to independent TCR β sequences. An independent TCR α chain or a TCR β chain sequence is NOT a receptor. Once this point is consistently clarified in the manuscript, your paper will be accepted.

We agree that our terminology was not specific enough and may confuse the readers. We have updated the manuscript to be more specific about single α or β chains and complete α/β TCR clonotypes. See our point-by-point response and implemented change below

Reviewer #1:Re-review:The major concern I have with the method, writing and the inability of this platform to provide a framework for prospective studies is the author's miss-statements (I assume they know better) that a TCR β chain sequence is not a TCR clonotype. Throughout the manuscript, they authors describe what they are analyzing as TCR clonotypes when they are actually referring to independent TCR β sequences. An independent TCRa chain or a TCRb chain sequence is NOT a receptor. The CDR3beta sequence does not equal or even mark a particular T cell clone.

We completely agree with the reviewer and have updated the manuscript to clarify this point.

We have edited the first paragraph of the Introduction:

“These receptors consist of two hypervariable chains, the α and β chains in the case of TCR. Progress in high throughput sequencing now allows for deep profiling of TCRalpha and TCRbeta chain repertoires, by establishing a near-complete list of unique receptor chain sequences, or ‘’clonotypes’’, present in a sample. Most sequencing data available corresponds to TCRbeta, but the same principles discussed below apply to TCRalpha repertoires or to paired α β repertoires.”

We also specified when we talk about single T-cell receptor chains specifically throughout the text.

Problem statement that this creates:Example in the Introduction: the authors argue that there are "Several mechanisms leading to the repertoire overlap. The first mechanism is convergent recombination…" This first statement or option is a product of sequencing just the TCRb chain, i.e., there are common rearrangements that occur do to V-J or V-D-J rearrangements that have no or few N-region additions. The second and third "options" are indeed TCRa + TCRb clonotypes, comparing the first possibility to 2 and 3 is comparing apples to oranges.Due to biases in V(D)J recombination process, the probability of generation of some receptors is very high, making them appear in almost every individual multiple times and repeatedly sampled in repertoire profiling experiments Britanova et al., (2014). This sharing does not result from a common specificity or function of the shared clonotypes and may in fact correspond to cells from the naive compartment in both donors Quigley et al., (2010), or from functionally distinct subsets such as CD4 and CD8 T-cells.The second possible reason for TCR sequence sharing is specific to identical twins, who may share T cell clones as a consequence of cord blood exchange in utero via a shared placenta Pogorelyy et al., (2017). The third and most interesting mechanism for sharing receptor sequences is convergent selection, in response to a common antigen.- The terminology makes these types of statements nonsensical. Identical TCR clonotypes will absolutely have the identical antigen-specificity.

We agree with the reviewer that case 2 (physical sharing of cells) implies the sharing of complete α/β TCR sequence. In case 3 (selection by affinity to the same antigen), the situation varies for different epitopes and in some cases tetramer-sorted T-cells of the same specificity have been reported to have the same TCRbeta but very different TCRalpha, and vice versa (see examples of paired sequences from (Dash et al., 2017), deposited to vdjdb.cdr3.net)). Therefore, the sharing of single receptor chain sequences due to selection by affinity to the same antigen is generally more likely than sharing of complete α/β TCR.

We have edited the second paragraph of the introduction to clarify these points in the following way:

“Comparison of sequenced repertoires has revealed that in any pair of individuals, large numbers of TCRbeta sequences have the same amino acid sequence Venturi et al., (2011). […] From functional studies, such as sequencing of MHC-multimer specific T-cells, it is known that the antigen-specific repertoire is often biased, and the same antigen-specific TCR β or α chain sequences can be found in different individuals Miles et al., (2011); Dash et al., (2017); Glanville et al., (2017).”

Issues that miss-representing TCRb sequences for TCR clonotypes in the general approach being promoted:The incorrect use of TCR clonotype leads to some particularly difficult to imagine scenarios. For examples, in the reviewer response and within the manuscript, "our method can be used to narrow down the potential candidates for further experimental validation of responsive receptors." (i.e., identify the antigen being recognized). Their idea regarding this approach is to use: "Functional tests (like cultivation with peptides, or cytokine secretion assays) are the ultimate way to confirm specificity of these predicted clonotypes."Within the cover letter: "We also are careful to note that this method should be used to identify antigen-specific candidates that need to be further verified by other methods. Nevertheless, we believe that identifying candidates for these experiments is extremely useful."

We agree with reviewer that the mere TCR β chain or the mere TCR α chain does not determine complete receptor specificity. The TCRbeta sequence is definitely not enough to determine receptor specificity in vitro. However, if a β-chain or an α-chain shared in patient cohort is identified as significant by our approach it is likely to be a part of T-cell receptor with the same specificity in each patient. T-cells from patients with the largest expansion of identified TCRbeta or TCRalpha clonotypes could then be used for single cell RNA sequencing to identify partner receptor chains or for various functional assays with candidate antigens.

We added sentences to the Discussion section on the contribution of both receptor chains to TCR specificity and possible assays to identify second receptor chain and possible antigen, see below.

- However, the sequencing and analyses method of the manuscript does not identify the V α chain, with only half of the receptor there is nothing to study further. Thus, there is no method provided that one could identify interesting receptors for prospective studies.

We agree that complete antigen-specific α/β T-cell receptor sequences would be much more informative and practically useful for prospective studies. Unfortunately, technologies for paired sequencing of α and β TCR chains (i.e. single-cell RNAseq) are still relatively low-throughput and expensive, comparing to the conventional RepSeq study targeting only single receptor chains. TCRalpha and TCRbeta pairing information is also impossible to retrieve from the bulk RNA-seq data. There are no published large datasets of complete paired α/β TCR sequences for cohorts of patients with the same condition. However, if such datasets were to become available in the future, our approach could be used directly to identify candidate complete receptor sequences there. The idea would be the same: the presence of shared TCR α chains, or β chains, or paired α β clonotypes with low recombination probability in many patients is the consequence of clonal expansion after recognition of the same antigenic peptide, and paired TCRbeta-TCRalpha datasets would allow us to retrieve the partner α chain (for TCRbeta condition-associated clonotypes, and vice versa for TCRalpha analysis) from each donor, yielding several complete α/β candidate receptors.

We also believe that even single condition-associated TCRbeta or TCRalpha chain sequences could be useful for diagnostic purposes, as was demonstrated by (Emerson et al., 2017) for CMV-associated TCRbeta clonotypes.

We have added the following text to the Discussion section to emphasize that single receptor chain does not fully determine TCR specificity. We also reference recently emerged methods to identify the second receptor chain, and to determine specificity of the complete α/β TCR, which might be useful in combination with our approach:

“Our approach can be used on other hypervariable receptor chains (TCR α, BCR heavy and light chains), as well as other species (mice, fish, etc.). […] As paired sequencing becomes more widespread, our method can be extended to the analysis of full paired TCR by applying the exact same analysis using the joint recombination probability of α β clonotypes.”

In summary, this is a method to identify outlier populations, for which retrospective data can be superimposed to create a snapshot of an immune response that has already been described using for example pMHC tetramers or other methods.